# Open Science Drone Toolkit: Open source hardware and software for aerial data capture

**Gustavo Pereyra Irujo** [ID] [1]*, **Paz Bernaldo**[2], **Luciano Velázquez**[3], **Antoni Pérez**[4], **Celeste Molina Favero**[5], **Alejandrina Egozcue**[6]

**1** Instituto Nacional de Tecnología Agropecuaria, Consejo Nacional de Investigaciones Científicas y Técnicas, Balcarce, Argentina, **2** Independent Researcher, Melipilla, Chile, **3** Facultad de Ciencias Agrarias, Universidad Nacional de Mar del Plata, Balcarce, Argentina, **4** Independent Researcher, Santiago, Chile, **5** Unidad Integrada Balcarce, Universidad Nacional de Mar del Plata, Instituto Nacional de Tecnología Agropecuaria, Balcarce, Argentina, **6** Independent Researcher, Balcarce, Argentina

* pereyrairujo.gustavo@conicet.gov.ar

## Abstract

Despite the increased access to scientific publications and data as a result of open science initiatives, access to scientific tools remains limited. Uncrewed aerial vehicles (UAVs, or drones) can be a powerful tool for research in disciplines such as agriculture and environmental sciences, but their use in research is currently dominated by proprietary, closed source tools. The objective of this work was to collect, curate, organize and test a set of open source tools for aerial data capture for research purposes. The Open Science Drone Toolkit was built through a collaborative and iterative process by more than 100 people in five countries, and comprises an open-hardware autonomous drone and off-the-shelf hardware, open-source software, and guides and protocols that enable the user to perform all the necessary tasks to obtain aerial data. Data obtained with this toolkit over a wheat field was compared to data from satellite imagery and a commercial hand-held sensor, finding a high correlation for both instruments. Our results demonstrate the possibility of capturing research-grade aerial data using affordable, accessible, and customizable open source software and hardware, and using open workflows.

## Introduction

Openness has been central to modern science since its inception [1], through the publication of theories and data on which they are based, and the encouragement of replication, scrutiny and challenge [2]. In recent decades, information technologies have enabled the rise of an "open science" global movement that seeks to not only increase transparency and dissemination of scientific processes and products, but also enable more widespread collaboration, participation and inclusion in science [3–5]. Although access to scientific publications and raw data has increased significantly in recent years [6–9], access to the scientific tools needed to obtain or analyze data (i.e., scientific instruments, materials and software) remains as one of the main barriers to increasing participation in science production, replication or reproduction of published results [10,11].

**Data Availability Statement:** Open Science Drone Toolkit details are publicly available from Vuela (https://vuela.cc/), and data for the example use

case is publicly available from the OSF repository (https://osf.io/t7y6x/).

**Funding:** Funding for this project was provided by Mozilla Foundation (Mozilla Science Mini Grant, foundation.mozilla.org) to GPI and PB, Knowledge/Culture/Ecologies (KCE2017, Western Sydney University, westernsydney.edu.au) to PB, Shuttleworth Foundation (Flash Grant, shuttleworthfoundation.org) to PB, Programa Cooperativo para el Desarrollo Tecnológico Agroalimentario y Agroindustrial del Cono Sur (PROCISUR, www.procisur.org.uy) to GPI, Instituto Nacional de Tecnología Agropecuaria (PNCYO-1124072 and 2019-PD-E3-I060, inta.gob.ar) to GPI. The funders had no role in study design, data collection and analysis, decision to publish, or preparation of the manuscript.

**Competing interests:** The authors have declared that no competing interests exist.

Open source software is computer code that is licensed so that the user has the freedom to copy and redistribute it, have access to the source code, and make improvements to it, among other rights [12]. Similarly, open source hardware is any physical object or artifact whose design is available so that anyone can study, modify, distribute, make, and sell the design or hardware based on that design [13]. Open source research software and open scientific instruments and materials are considered to provide a series of advantages over proprietary alternatives: i) being either cost-free (in the case of software) or usually more affordable (in the case of hardware), they allow more people to participate in science endeavors, especially for non-professional or budget-limited researchers; ii) reproducibility of published results or replication attempts are less constrained by a lack of access to the same tools that were originally used; iii) having access to the software code or hardware design allows for a better understanding of the functioning of the tool and the methods or algorithms that it implements; and iv) it is possible to customize the tools to adapt them to new uses or local contexts [11,14–16].

Uncrewed aerial vehicles (UAVs, usually called "drones") can be a powerful tool for research in disciplines such as agriculture and environmental sciences, allowing the capture of high-resolution aerial imaging with great speed and flexibility [17]. Drone use in research is rapidly growing but it is dominated by closed source tools: in a recent literature review of applications in agro-environmental monitoring, more than 80% of studies used fully closed-source drones, and more than 90% of studies used proprietary closed-source software for image processing [18]. Proprietary drone solutions usually require a significant initial investment, monthly software subscriptions, or an internet connection for cloud processing, which can constitute barriers for many low-resource users [19,20], and usually function as a 'black-box' which offer users little insight on its internal workings, and limited customization [21]. Also, when these solutions are implemented in developing countries, concerns have been raised regarding limited repairability, and risks of technological dependence and extractive practices [22].

For a typical use of drones in environmental or agricultural research, the drone needs to be able to be reliably and precisely positioned over the studied terrain and capture images that can be later processed to get a high quality image of the surveyed area and extract useful data [23–25]. Although there are already many open source hardware and software tools that can be used in each of the individual steps of this process, our question was whether it was possible to perform all these steps using open tools. The objective of this work was to address this question by collecting, curating, organizing and testing a comprehensive set of open source tools for aerial data capture for research purposes. The result of these actions is the Open Science Drone Toolkit (OSDT), which is presented here in detail, and is also available online at https://vuela.cc/en/toolkit.

## Toolkit design process

The OSDT was developed as part of "Vuela" [26,27], a research-action project which aimed to fight the lack of access to creating scientific and technological knowledge, by exploring an alternative way of developing scientific tools. The toolkit was built through a collaborative and iterative process, involving the work of more than 100 people between 2017 and 2019, in more than 30 local, in-person workshops, in five countries (Argentina, Brazil, Chile, Paraguay and Uruguay), as well as permanent online collaboration. Workshop participants were school students, traditional scientists, technicians, hobbyists, journalists, local community members, self-taught software developers, and included both academics and people with no formal academic or technology background, and people with and without experience making or using

**Table 1. Tasks that can be performed using the Open Science Drone Toolkit in order to obtain data from aerial images.**

| Task | Details |
|---|---|
| **0. Assemble the toolkit** | Build or acquire the drone, camera and other hardware components, install the software components, and perform the necessary configurations, following the toolkit assembly guide. |
| **1. Identify the study area** | Specify the boundaries of the area that will be surveyed, either in situ or using georeferenced aerial or satellite imagery |
| **2. Design the flight plan** | Design the flight path the drone will follow over the study area, as well as flight altitude and speed, taking into account the desired ground sampling resolution (cm per pixel), the overlap required for image stitching, and technical constraints of the drone and camera |
| **3. Select the camera settings** | Select the optimal camera parameters (shutter exposure, ISO sensitivity, automatic shutter interval) according to current conditions at the time of the flight |
| **4. Perform a satellite-guided flight** | Perform a satellite-guided autonomous flight over the study area, following the designed flight plan, capturing the necessary images with a fixed set of camera parameters |
| **5. Geotag the captured images** | Assign coordinates to each image according to the precise location recorded by the drone at capture time |
| **6. Process the images to obtain a mosaic** | Process the captured images in order to obtain an orthorectified and georeferenced image of the complete surveyed area |
| **7. Analyze the mosaic to obtain data** | Process the mosaic image in order to obtain data relevant to the research question, for the complete image or for specific areas |
| **8. Manage and share data** | Organize and visualize all the generated data (raw images, mosaic, area boundaries, flight plan and telemetry records) and metadata (flight name and description, date and time, location, etc.), and bundle all files together for efficient archiving and sharing |

drones. Work was carried out also in different languages: Spanish, Haitian Créole, French, Portuguese and English.

The Open Science Drone Toolkit is a set of hardware and software tools, guides, and protocols that enable the user to perform all the necessary tasks to obtain aerial data, as detailed in Table 1. These steps represent a 'typical' use case, but can be modified according to the specific research question.

One of the first objectives of the project was to put into practice one of the commonly less-exercised freedoms of open source hardware: the freedom to modify an existing design. Instead of developing a drone from scratch, the project started by replicating, testing, and identifying potential improvements for an already available open-source drone called "Flone" [28]. The original design had limited capabilities for performing the required tasks listed in Table 1, so a series of changes were needed for using the drone for research purposes: increasing the range that the drone could safely cover, adding satellite navigation capability, increasing the payload capacity, and improving the stability of the camera. This iterative process of hardware development was carried out in conjunction with a careful selection of open source software tools (and development of new ones) to perform each of the tasks listed in Table 1, and the development of protocols and detailed user guides. The resulting set of tools is described in the following section.

## Toolkit components

The components of the OSDT are listed in Table 2. The main component is the open-hardware drone, developed especially for this toolkit. Other hardware components of the toolkit are not open source, but are mostly off-the-shelf equipment that can be readily replaced. The software components of the toolkit are all open-source software projects, which have been selected for

Table 2. Summary of the hardware, software, and documentation components of the Open Science Drone Toolkit.

| Component type | Component | Task in which is used |
|---|---|---|
| **Hardware** | Computer (generic PC with Windows operating system) | 0 to 8 |
| | Smartphone (generic smartphone with Android operating system) | 1 |
| | Camera (Canon brand camera compatible with CHDK software) | 3, 4 |
| | Drone (OVLI drone) | 4 |
| | Radio transmitter (generic 6-channel radio transmitter) | 4 |
| | Batteries and charger (generic 3-cell lithium-polymer battery and balance charger) | 4 |
| **Software** | Location recorder app (GPS Logger) | 1 |
| | Drone ground station (Mission Planner) | 2, 4, 5 |
| | Camera control (CHDK) | 3, 4 |
| | Drone autopilot (ArduCopter) | 4 |
| | Image processing (OpenDroneMap) | 6 |
| | Orthomosaic processing and analysis (QGIS) | 7 |
| | Data management (Bitácora) | 8 |
| **Documentation** | Toolkit assembly guide | 0 |
| | Toolkit usage guide | 1 to 8 |

being suitable for each task, or developed especially for the toolkit. Finally, the documentation includes assembly and usage guides for the toolkit.

The OVLI drone (an acronym for "Objeto Volador Libre", which means "Free Flying Object" in Spanish) is a quadcopter (i.e., a helicopter with four propellers), equipped with an autopilot board with accelerometer, gyroscope, barometer, and GNSS (Global Navigation Satellite System) sensors that allow fully autonomous flight. The autopilot is an open source Pixhawk board [29,30], running the open source ArduPilot/ArduCopter firmware [31,32]. The OVLI has a frame diameter of 395mm and weighs 0.773kg without batteries. Its frame is assembled from MDF (Medium Density Fibreboard) cut with a laser cutter according to a design file, which can be easily edited to modify the drone structure. This material was chosen because it is widely available, low cost, and easy to assemble, repair and modify. The final design of the OVLI is shown in Fig 1, alongside the original "Flone" drone on which it was based.

The OVLI drone has a payload capacity of around 500 g, enough for a high-resolution RGB (Red-Green-Blue, *i.e.*, visible spectrum) camera, multi-spectral camera, or other sensors. Maximum flight time is 11 minutes, using a 5000 mAh battery and carrying a 141 g camera as payload, measured from take-off until the low-battery alarm was activated, with approximately 30% of battery capacity remaining.

Operation of the OVLI drone can be done both through a manually operated radio controller, but for research purposes it is usually convenient to fly the drone autonomously using a pre-programmed flight plan. Planning usually begins by identifying the area to be surveyed (task #1 in Table 1), which can be done by physically surrounding the area carrying a smartphone and using the open-source app GPSLogger (Table 2). The resulting file with the coordinates is uploaded to the open-source software Mission Planner (Table 2) to design the flight plan (task #2 in Table 1), according to the desired image resolution, and considering the maximum flight time of the drone and other constraints. This flight plan is uploaded to the OVLI drone, which can later fly autonomously while capturing the images (task #4 in Table 1). During flight, the Mission Planner software is also used to view live telemetry data, such as the position of the drone, battery voltage, altitude, speed, etc.

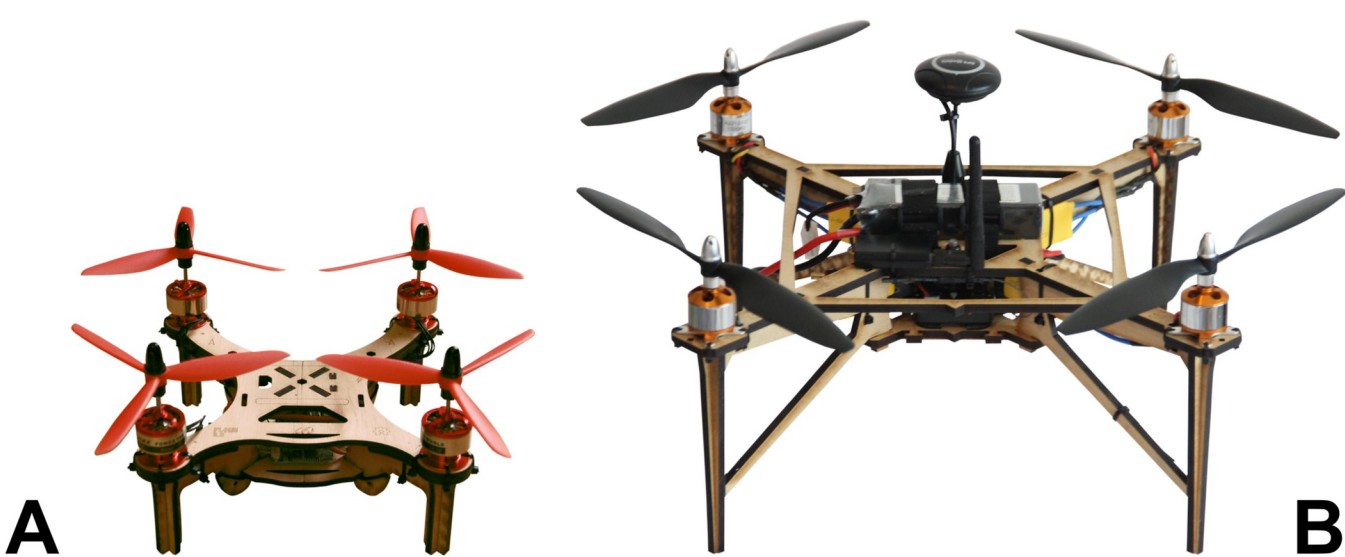

**Fig 1. The Flone and OVLI drones.** (A) original "Flone" design [28] upon which further versions of the open hardware drone were developed. (B) the "OVLI" drone that was developed as part of the Open Science Drone Toolkit.

The camera selected for the toolkit is an RGB, 12-megapixel pocket camera (Canon Power-Shot ELPH100HS, or equivalent). This kind of camera was selected for two main reasons: 1) "pocket" or "point and shoot" cameras usually have a mechanical shutter, which means that when an image is captured, all the pixels are captured at the same time, whereas many "action cameras" usually used in drones have an electronic "rolling shutter", in which the sensor captures the images line-by-line, potentially introducing image distortions [33]; 2) most Canon cameras have the possibility of being "hacked" by means of the open-source CHDK (Canon Hack Development Kit) software [34], which allows setting the camera to capture images automatically, and to manually set camera parameters (e.g., shutter speed and ISO values; task #3 in Table 1) to capture sharp, well-exposed and suitable images for further processing and data extraction [35]. This type of camera does not provide location data of the images (which is necessary for later obtaining a georeferenced mosaic), so this information has to be retrieved from the flight log of the drone. This process is called geotagging (task #5 in Table 1), and can be performed using the Mission Planner ground station software. This kind of off-the-shelf RGB cameras have been shown to be useful for the measurement of vegetation indices [36]. Moreover, the camera filters can be modified by replacing the standard near-infrared (NIR) filter with an appropriate filter, resulting in a low-cost camera capable of detecting two visible bands and one NIR band [37,38].

The captured images then need to be merged to obtain a rectified and georeferenced image of the complete surveyed area, which is known as an orthomosaic (task #6 in Table 1). Currently, a suitable open-source software to perform this task is OpenDroneMap [39], which has been shown to provide high quality results comparable to widely-used commercial packages [21]. The next step is to extract information from the orthomosaic image. This step depends largely on the research question that is being addressed. The open source software QGIS [40] can be used to open the georeferenced orthomosaic and calculate vegetation indices, measure areas, and multiple other data extraction tasks (task #7 in Table 1).

The open-source application "Bitácora" ("logbook" in Spanish) (https://vuela.cc/en/ bitacora), developed especially for the OSDT, helps in visualizing and organizing all the files, images and metadata generated during the whole process, for archiving, sharing or further

processing. The user only needs to save all the files generated in a flight (survey area polygon, flight plan, captured images, mosaic, elevation model, etc.) in a folder, and the program will automatically generate a map visualizing the files, and a table with flight information (flight date and time, location, altitude, speed, names of relevant files; Fig 2). This information is also saved in open formats compatible with other software (flight information table in csv format, flight map in png and kml formats).

The toolkit documentation includes: 1) an "assembly guide" that includes a step-by-step guide for building the OVLI drone, setting up and configuring the hardware components, and installing the software, and 2) a "usage guide", with instructions for flying the drone, programming an autonomous mission, programming the camera, and processing the images. Both are available as openly licensed documents (using a Creative Commons license allowing users to redistribute and build upon the material, with attribution), ready for download in PDF and HTML format, and also as live documents (in Google Docs) open for suggestions. The guides have a simple layout that allow for automatic machine translations in many languages, which are readily available from the project website.

## Example use case

An example use case is presented here in which the OSDT was used to obtain data on the spatial variability in the maturity of a wheat crop, assessed through a vegetation index that quantifies the "greenness" of the crop canopy. A 6000 m$^2$ area was surveyed in a wheat field sown in

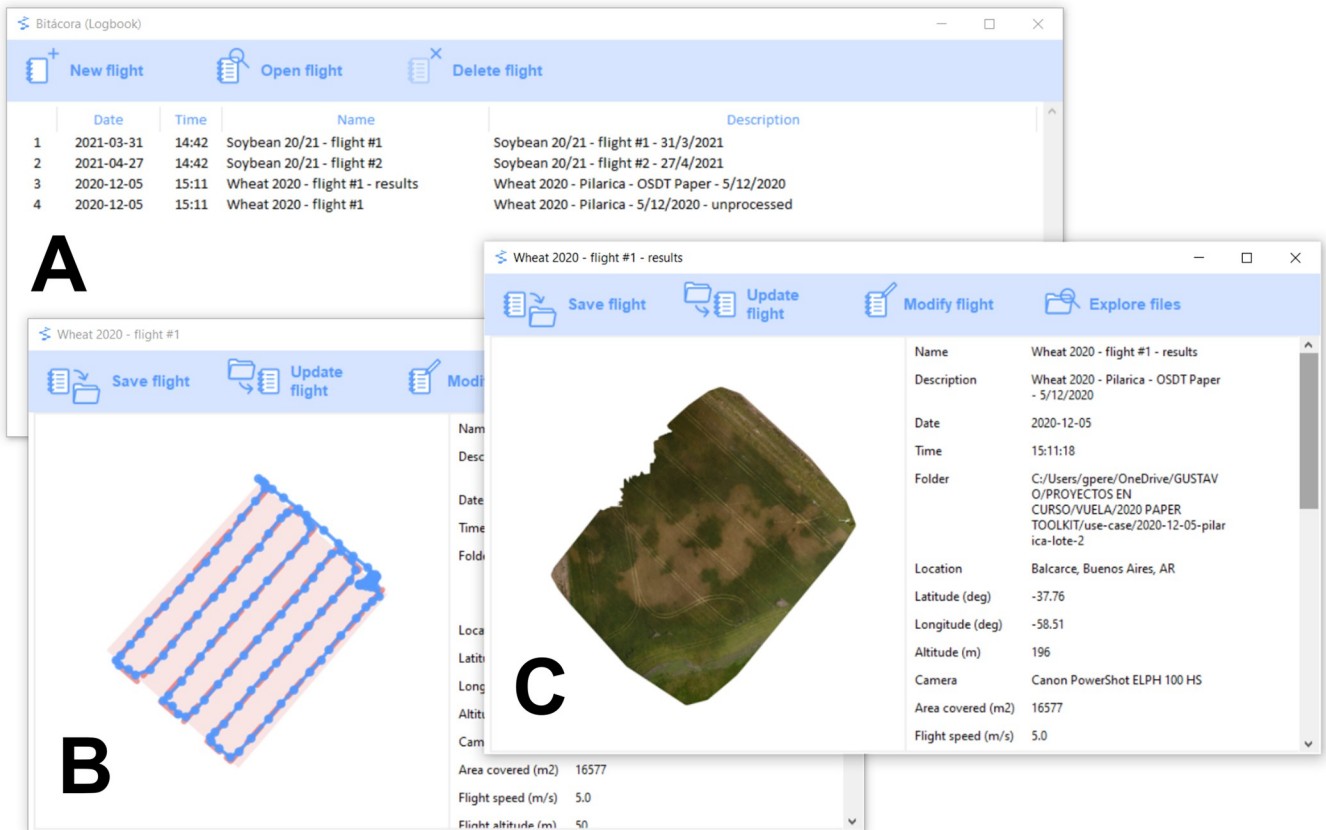

**Fig 2. Screenshot from the "Bitácora" software.** (A) The main window, showing the list of logs already registered in the logbook. (B-C) Individual flight windows showing details for selected flights.

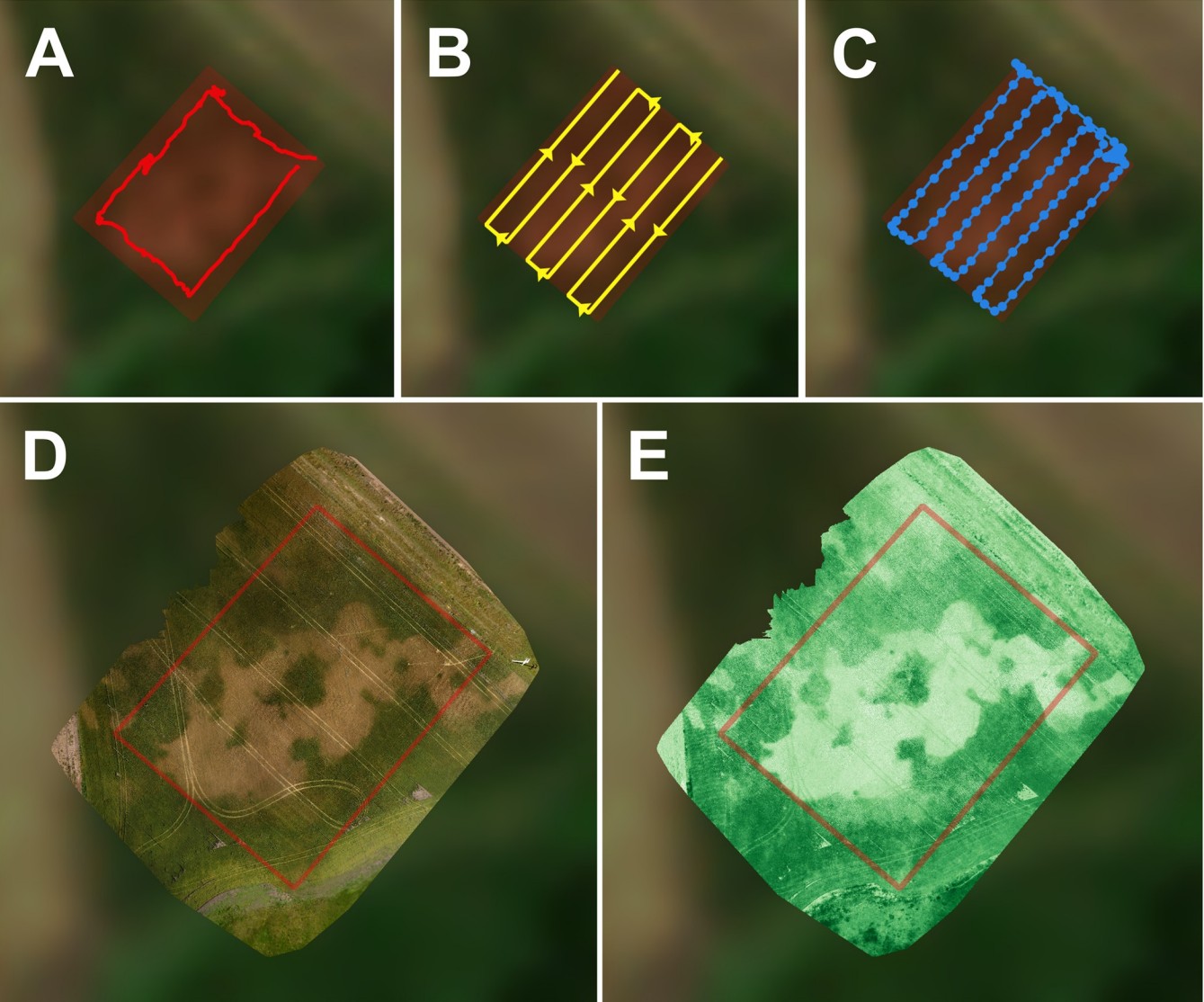

**Fig 3. Aerial data capture and analysis steps followed in the presented example use case.** A) identification of the study area, which was done first *in situ* with a location recorder app (bright red line) and then manually refined by drawing a rectangular polygon (dark red area); B) design of the flight plan to cover the study area using a "grid" or "lawn-mower" pattern; C) flight and image capture, represented through the actual flight path of the drone (blue line) and the position of each captured image (blue circles); D) orthomosaic obtained by joining the captured images; and E) calculated vegetation index (VARI) calculated from the orthomosaic data. The background image is a 10x10m resolution satellite image captured 1 day after the drone flight (retrieved from Sentinel Hub EO Browser under a CC-BY 4.0 license [45]).

late July 2020 in Balcarce, Buenos Aires Province, Argentina. This particular field was selected for having uneven maturity of the crop due to variability in soil depth. Following the steps in the toolkit guide, the area of interest was first delimited (Fig 3A), a flight plan was designed (Fig 3B), and the OVLI drone was flown on the 5th of December 2020, when the crop was in the grain filling stage (Zadoks 7.7 stage). A total of 151 images were captured at a flight altitude of 50m (Fig 3C), and 104 of them (discarding those captured during takeoff and landing) were then used to obtain an orthomosaic with a resolution of 2 cm/pixel (Fig 3D).

Vegetation indices are transformations of data obtained from optical sensors, usually based on the plant's increased reflectance in the green and/or infrared wavelengths, that can be used

to quantify spatial and temporal variations in vegetation characteristics [41]. A vegetation index (Visible Atmospherically Resistant Index, VARI) that has been previously used to estimate wheat growth and phenology [42,43] was calculated from the red, green and blue channels of the orthomosaic, in order to quantify the "greenness" of the crop as an indicator of the degree of maturity (Fig 3E). This data obtained from the drone images was compared to the Normalized Difference Vegetation Index (NDVI), which is the most widely used vegetation index [44], obtained from two sources: a handheld sensor and satellite data.

Satellite data from the Sentinel 2 constellation (European Space Agency) was retrieved from the Sentinel Hub EO Browser under a CC-BY 4.0 license [45]. Data from bands B04 (red, central wavelength = 665nm) and B08 (near infrared, central wavelength = 842nm) for the 6th December 2020 (1 day after the drone image capture) at 10m/pixel resolution was used to calculate NDVI values for the surveyed area (Fig 4A). For comparison between satellite (NDVI) and drone (VARI) data, 140 10 x 10 m areas equivalent to the satellite image pixels were delimited in the processed orthomosaic (Fig 4C), and the mean VARI value measured in each of them. One caveat to this comparison was that the georeferencing of the orthomosaic was based on the drone GNSS sensor which usually has an accuracy of about 2 m [46], so the correspondence between these 10 x 10 m areas and the satellite image pixels might not have been complete.

On the 28th of November 2020 (seven days before the drone image capture) a hand-held sensor (Greenseeker, N-tech Industries, USA) was used to measure NDVI in six transects parallel to the crop rows, each of them 70 meters long and spaced 18 meters apart (Fig 4D). To aid in the correspondence between sensor and drone data, a visible mark that could be easily identified in the aerial images was placed at the start of the first transect, and each subsequent transect was started based on ground measurements relative to that reference. The sensor was placed one meter above the canopy, resulting in a field of view of about 60 cm. Around 700 data points were recorded in each transect, equivalent to around one point every 10 cm. For comparison between handheld NDVI sensor and drone (VARI) data, NDVI data was averaged every two meters (yielding 35 data points per transect, and 210 in total). Similarly, 35 2 x 2 m areas along each transect were delimited in the processed orthomosaic, and the mean VARI value measured in each of them.

VARI data obtained from the drone orthomosaic were plotted against NDVI data from the satellite image and the handheld sensor, and an exponential equation of the form $y = ax^b - c$ was fitted for each set of data. The NDVI index usually has a curvilinear relationship with the green leaf area index, while the VARI index has been shown to have a rather linear relationship [42], therefore a curvilinear relationship between NDVI and VARI can be expected. A high correlation was found in both cases, with $R^2$ values of 0.84 and 0.88 when drone data was compared to satellite and hand-held sensor data, respectively (Fig 4E and 4F). A single curve could also be fit to both datasets, with an $R^2$ of 0.86 (not shown).

## Discussion

This paper reports the results of developing a complete toolkit in order to demonstrate the possibility of capturing research-grade aerial data using open source software and hardware. The use case example presented shows its suitability for tasks that can be useful for many research questions (and also commercial applications, *e.g.*, farming). The wide range of possible applications of aerial imaging in research, however, cannot be fully covered by any single toolkit. Nevertheless, the open nature of the OSDT allows for its components to be used separately or be replaced by alternative tools, either open-source or proprietary, as necessary. For example, if the area of interest is significantly larger than the one shown in the example, the flight time

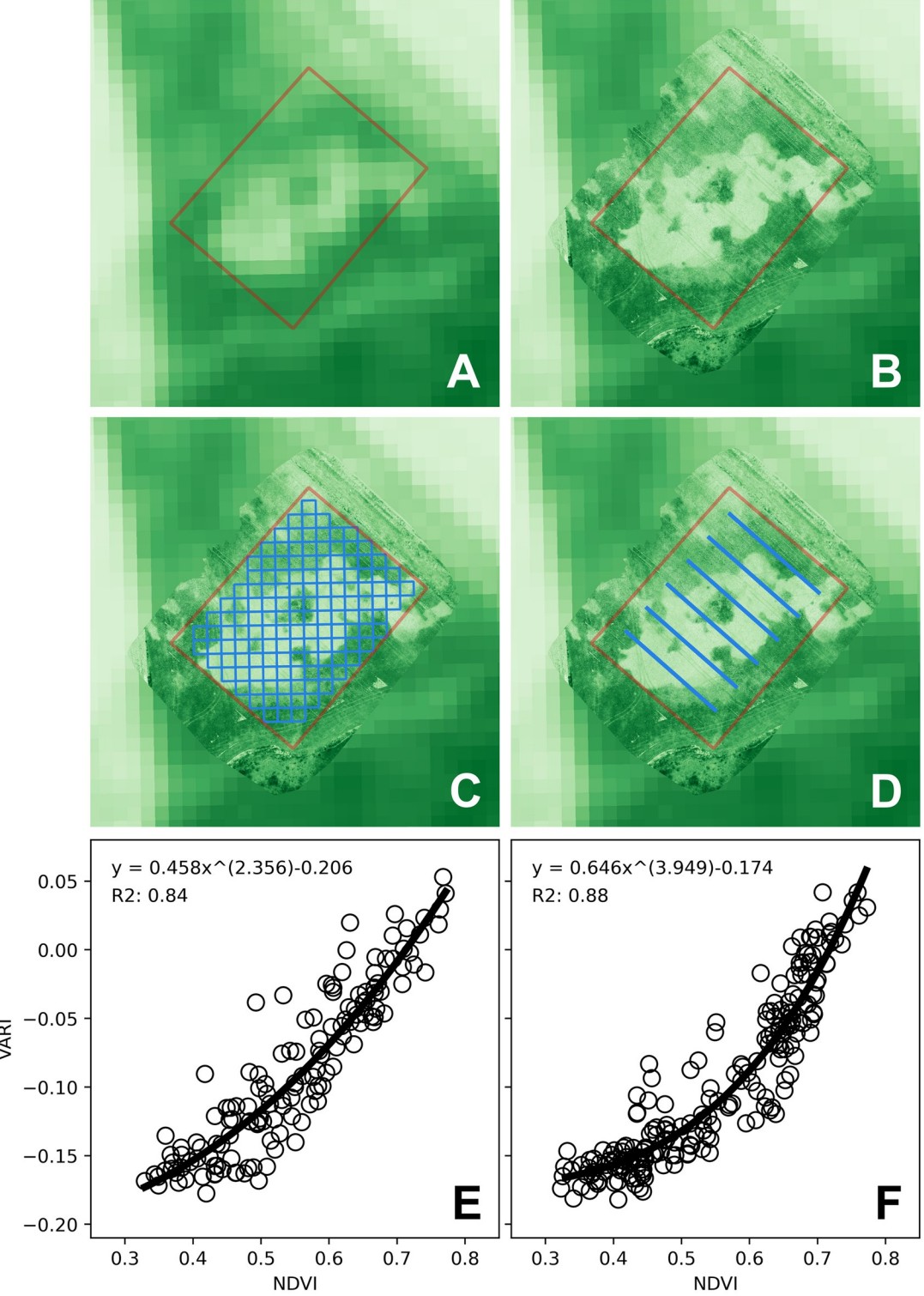

**Fig 4. Comparison between data obtained with the OSDT and data obtained from satellite imagery and a handheld sensor.** A) NDVI satellite image from Sentinel 2; B) VARI image from drone orthomosaic; C) VARI drone image showing the 10x10m areas equivalent to the satellite image pixels delimited in the processed orthomosaic; D) transects used for measurement with hand-held sensor; E) relationship between NDVI from the satellite image and VARI from the drone; and F) relationship between NDVI from the hand-held sensor and VARI from the drone. Satellite image data was retrieved from Sentinel Hub EO Browser under a CC-BY 4.0 license [45].

of the OVLI drone would not be sufficient. In that case, a "fixed wing" type of drone, such as the open-source "Asa-Branca-I" [47], could provide the capability of covering more than 100 ha in a single flight (albeit with lower image resolution). And if the cost of a drone is a limitation, or if drones cannot be used due to regulations, aerial images can also be captured using kites [48]. It is also possible to use the image processing, analysis and management software tools of the toolkit with aerial images obtained with proprietary drones.

Concerns about the quality and reliability of open source tools (especially hardware) can sometimes limit their use [49]. In this work we attempted to overcome this limitation by comparing the results obtained with the OSDT to data from satellite imagery and a commercial hand-held sensor, finding a high correlation for both instruments. The two sensors used for comparison have been among the most popular tools used by farmers for crop monitoring for around 20 years [50], both of them make use of the most widely used vegetation index (NDVI) [44], and can therefore be considered as a reliable benchmark. Studies with off-the-shelf RGB cameras such as that used in the OSDT have shown that they can yield robust measurements of vegetation indices [36], which can also be further improved through radiometric calibrations [51], with results comparable to multispectral cameras which can cost several times more [52–55], and with higher spatial resolution [56]. Although not shown in this paper, the OSDT can also be used to generate digital elevation maps and 3D point clouds of the studied terrain. Studies using similar tools showed that it is possible to obtain results similar to those of expensive LiDAR (Light Detection And Ranging) systems, when appropriate ground control points are used [57,58].

Another concern about open source research tools is their sustainability [59], especially when they lack an associated business model to provide funding, considering that long-term availability is important for reproducibility and replicability [60]. Nevertheless, open source is usually considered as a requisite for sustainable research software, as it ensures the possibility of continuous validation, reuse and improvement [61,62] while, on the other hand, proprietary software has been found to be an obstacle for reproducibility [63]. The issue of sustainability of open hardware has not been studied extensively [64], but it has been argued that open source hardware can provide more long-term security to research projects due to the possibility of in-house repairing in case the original provider went out of business [14]. We therefore argue that using the OSDT for aerial data capture for research purposes could be considered a more sustainable option than proprietary commercial systems.

While there have been significant technical improvements in drones and sensors in recent years, little attention has been paid to the management and storage of the increasingly large and complex datasets that are the result of drone operations [65]. There are few standards for drone data management, sharing or publication, which makes collaboration and reproducibility difficult [66]. Commercial drone packages usually offer complete solutions, but generally at the cost of expensive licenses and less interoperability with other tools or possibility of customization. Open source tools offer a more flexible but fragmented landscape, and users can be set back by the need to deal with many individual components. In the OSDT, the software "Bitácora" was developed with the aim of helping to overcome this issue, by providing a way of centralizing all the files and data that is generated in the different steps of the process of aerial data capture, which would otherwise have to be managed and visualized using many different tools. As an open source software, it can also be extended to incorporate other data formats and file types (e.g. flight plans for proprietary drones, images from multispectral cameras, etc.). It also aids the user in collecting all the generated data in a single folder, adding the corresponding metadata, which helps in openly sharing research data in reusable and interoperable formats. "Bitácora" can automatically extract part of the metadata required in the Minimum Information Framework for drone users (proposed by [66]), thus helping users make their data

"FAIR" (findable, accessible, interoperable and reusable). Another open source software with a similar goal is "DroneDB" [67], which aids especially in sharing datasets of images, orthomosaics and other drone products through a cloud interface, but without providing flight metadata. "Bitácora" is therefore a small but key component of the OSDT, since it helps "bundle" the toolkit together and use it to further the goals of open science.

## Conclusions

The Open Science Drone Toolkit was presented in this paper, which enables the user to perform all the necessary tasks to obtain aerial data. Data obtained with this toolkit over a wheat field was compared to data from satellite imagery and a commercial hand-held sensor, finding a high correlation for both instruments. These results demonstrate the possibility of capturing research-grade aerial data using affordable, accessible and customizable open source software and hardware, and using open workflows.

## Supporting information

**S1 File. Alternative language article.** Complete manuscript in Spanish: "Herramientas de código abierto para la captura de datos aéreos mediante drones".
(PDF)

## Acknowledgments

We would like to acknowledge the contributions of the participants of the Vuela project workshops and online collaborators that made building this toolkit possible, especially Loulou Jude, Daniela Muñoz, Lot Amorós, Guillermo Pereyra Irujo, Nicolás Narváez, Carla Alvial, Fernando Yamada, John Arancibia, Constanza Alberio, Vicente Dimuro, and Stevens Azima. We want to thank Junta de Vecinos Teniente Merino Alto, Junta de Vecinos Francisco Werchez, Universidad Católica, INIA Rayentué (Chile), INTA Balcarce, Club Social de Innovación Balcarce, R'lyeh Hacklab, INTA Marcos Juárez, and Universidad Nacional de Cuyo (Argentina), IPTA Capitán Miranda (Paraguay), Universidade Federal do Rio Grande do Sul (Brazil), and INIA La Estanzuela (Uruguay) for providing the venues for the workshops. We also want to thank Abril Pereyra Molina and Julián Pereyra Molina for their assistance with field measurements.

## Author Contributions

**Conceptualization:** Gustavo Pereyra Irujo, Paz Bernaldo, Luciano Velázquez, Antoni Pérez, Alejandrina Egozcue.

**Data curation:** Gustavo Pereyra Irujo, Celeste Molina Favero.

**Formal analysis:** Gustavo Pereyra Irujo.

**Funding acquisition:** Gustavo Pereyra Irujo, Paz Bernaldo.

**Investigation:** Gustavo Pereyra Irujo, Paz Bernaldo, Luciano Velázquez, Antoni Pérez, Celeste Molina Favero, Alejandrina Egozcue.

**Methodology:** Gustavo Pereyra Irujo, Paz Bernaldo, Luciano Velázquez, Antoni Pérez, Alejandrina Egozcue.

**Project administration:** Gustavo Pereyra Irujo, Paz Bernaldo.

**Resources:** Gustavo Pereyra Irujo, Paz Bernaldo, Antoni Pérez, Alejandrina Egozcue.

**Software:** Gustavo Pereyra Irujo, Paz Bernaldo, Luciano Velázquez, Antoni Pérez, Alejandrina Egozcue.

**Supervision:** Gustavo Pereyra Irujo, Paz Bernaldo, Luciano Velázquez, Antoni Pérez, Celeste Molina Favero, Alejandrina Egozcue.

**Validation:** Gustavo Pereyra Irujo, Paz Bernaldo, Luciano Velázquez, Antoni Pérez, Celeste Molina Favero.

**Visualization:** Gustavo Pereyra Irujo.

**Writing – original draft:** Gustavo Pereyra Irujo.

**Writing – review & editing:** Gustavo Pereyra Irujo, Paz Bernaldo, Luciano Velázquez, Antoni Pérez, Celeste Molina Favero, Alejandrina Egozcue.

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
