## [Decision Letter · Decision Letter 0]

18 Jan 2023

PONE-D-22-24873Open Science Drone Toolkit: open source hardware and software for aerial data capturePLOS ONE

Dear Dr. Pereyra Irujo,

Thank you for submitting your manuscript to PLOS ONE. First, I would like to apologize for the delay for the reviewing. It is not due to the qualtity of your work, but the lack of available expert to review it.After careful consideration, we feel that it has merit but does not fully meet PLOS ONE’s publication criteria as it currently stands. Therefore, we invite you to submit a revised version of the manuscript that addresses the points raised during the review process.

We look forward to receiving your revised manuscript.

Kind regards,

Vona Méléder, Ph.D.

Academic Editor

PLOS ONE

and https://journals.plos.org/plosone/s/file?id=ba62/PLOSOne_formatting_sample_title_authors_affiliations.pdf.

3. We note that [Figures 2-4] in your submission contain [map/satellite] images which may be copyrighted. All PLOS content is published under the Creative Commons Attribution License (CC BY 4.0), which means that the manuscript, images, and Supporting Information files will be freely available online, and any third party is permitted to access, download, copy, distribute, and use these materials in any way, even commercially, with proper attribution. For these reasons, we cannot publish previously copyrighted maps or satellite images created using proprietary data, such as Google software (Google Maps, Street View, and Earth). For more information, see our copyright guidelines: http://journals.plos.org/plosone/s/licenses-and-copyright.

a. You may seek permission from the original copyright holder of Figures 2-4 to publish the content specifically under the CC BY 4.0 license. 

Natural Earth (public domain): http://www.naturalearthdata.com/.

Reviewers' comments:

Reviewer's Responses to Questions

**Comments to the Author**

1. Is the manuscript technically sound, and do the data support the conclusions?

Reviewer #1: Partly

Reviewer #2: Yes

2. Has the statistical analysis been performed appropriately and rigorously? 

Reviewer #1: I Don't Know

Reviewer #2: Yes

3. Have the authors made all data underlying the findings in their manuscript fully available?

Reviewer #1: Yes

Reviewer #2: Yes

4. Is the manuscript presented in an intelligible fashion and written in standard English?

Reviewer #1: Yes

Reviewer #2: Yes

5. Review Comments to the Author

Reviewer #1: 1. This is a valuable project that will be of interest, and I believe a revised version should be published; my two main recommendations are: a) carefully address reproducibility and replicability (R&R) literature, and b) do not try to “validate” data collected from the platforms but provide comparisons

2. Abstract; given the relatively large altitude of an orbital platform, it is not clear why satellite imagery would be the most obvious modality for reference

3. Abstract; reproducibility is mentioned as a benefit of the Open Science Drone Toolkit, though this is presented as a conlcusion rather than an area of inquiry; how does one ascertain reproducibility and what about replicability and other stakeholder interests?

4. Line 36; it is not clear what the authors mean when using the word “reproduction”, which has limited reference in The Royal Society (2012). The National Academies (2019) provides a much needed clarification on terminology regarding reproducibility and replicability, with the latter being mentioned far more often by The Royal Society (2012): National Academies of Sciences, Engineering, and Medicine. (2019). *Reproducibility and replicability in science*. National Academies Press. [https://www.nap.edu/catalog/25303](https://www.nap.edu/catalog/25303); these definitions and their implications would need to be addressed throughout the manuscript in order to avoid confusion with recent progress on reproducibility and replicability (R&R)

5. Line 44; if the user makes improvements to open source software, as described here, does that contribute to (or detract from) its benefits for others?

6. Lines 100-102; the statement that the hardware which is not open-hardware can be “readily replaced” doesn’t make sense in the context of the study; if the various commercial components such as Windows computers are convenient, why not commercial drones?

7. P. 6; when referencing “GPS”, this would typically refer to Global Positioning System which is a United States platform; wouldn’t this benefit from a more generic reference, given international scope?

8. Line 185; I doubt Sentinel 2 imagery can be used to “validate” drone imagery that has a spatial resolution of 2 x 2 cm; these are very different sources and it is not clear what is meant by “validate”; I would strongly suggest including these efforts as a comparison but not as a validation

9. The use of the handheld sensor is helpful, though still a comparison rather than a validation; I believe the latter would require a far more sophisticated process which draws deeply upon literature regarding error expression and propagation as well as uncertainty

10. You can reference literature that has already done extensive work on low altitude drone imaging using commercial sensors, and its comparisons with reference data; the fact that you are using an open hardware drone should not change the fundamental process of low altitude aerial image streams

Reviewer #2: Thanks for the opportunity to review this paper, I really enjoyed reading it an learning about your work! The paper is well written and provides documentation of interesting science that is well aligned with my area of interest. I have attached my comments in the document and hopefully they prove useful for you in a minor revision.

I also suggest that the figure quality needs to be increased - they are quite blurry on my screen.

6. PLOS authors have the option to publish the peer review history of their article (what does this mean?). If published, this will include your full peer review and any attached files.

Reviewer #1: No

Reviewer #2: **Yes: **Karen Joyce

---

## [Author Response · Author response to Decision Letter 0]

16 Feb 2023

Review Comments to the Author - Reviewer #1

Reviewer's comment: 1. This is a valuable project that will be of interest, and I believe a revised version should be published; my two main recommendations are: a) carefully address reproducibility and replicability (R&R) literature, and b) do not try to “validate” data collected from the platforms but provide comparisons

Authors' response: We appreciate the reviewer’s comments and valuable suggestions. We have addressed both of the main recommendations, as detailed in the responses to the following comments.

Reviewer's comment: 2. Abstract; given the relatively large altitude of an orbital platform, it is not clear why satellite imagery would be the most obvious modality for reference

Authors' response: Both satellite imagery and hand-held sensors (such as the one used in our study) are among the tools most frequently used by farmers to monitor their crops, have been in use for around 20 years, and therefore can safely be considered as well-established, trusted, reference sensors (we included a comment and reference in the Discussion section to clarify this, lines 250-252 *). An “obvious” choice could have been another UAV, but none was available to us at the moment, and also the myriad of currently available platforms and sensors would have made it difficult to select one of them as reference (the citations added in line 255 [52-55] include many of those sensors, detailed in our response to Comment #10).

* Line numbers mentioned in our responses correspond to the revised version with tracked changes

Reviewer's comment: 3. Abstract; reproducibility is mentioned as a benefit of the Open Science Drone Toolkit, though this is presented as a conclusion rather than an area of inquiry; how does one ascertain reproducibility and what about replicability and other stakeholder interests?

Authors' response: As indicated previously, R&R terminology has been clarified, and therefore the reference to reproducibility in its generic sense was removed from the Abstract and the Conclusions. Our work does not attempt to evaluate the reproducibility of our computational methods nor the replicability of the results obtained with the OSDT, as it would be out of the scope of the paper. We have therefore revised the text and removed any references to the reproducibility or replicability of the methods and results of our paper. In the Introduction section, however, several references are cited which argue that open source hardware and software can be beneficial for R&R. These references were kept, but we clarified the terminology as per the NASEM (2019) report (details in the response to the following comment). 

Reviewer's comment: 4. Line 36; it is not clear what the authors mean when using the word “reproduction”, which has limited reference in The Royal Society (2012). The National Academies (2019) provides a much needed clarification on terminology regarding reproducibility and replicability, with the latter being mentioned far more often by The Royal Society (2012): National Academies of Sciences, Engineering, and Medicine. (2019). *Reproducibility and replicability in science*. National Academies Press. [https://www.nap.edu/catalog/25303](https://www.nap.edu/catalog/25303); these definitions and their implications would need to be addressed throughout the manuscript in order to avoid confusion with recent progress on reproducibility and replicability (R&R)

Authors' response: The use of R&R terminology was revised throughout the manuscript, following the NASEM (2019) report, as detailed here:

Lines 31-32: “reproducible” was removed (open source software and open data provide increased transparency which could arguably contribute to reproducibility, but the reproducibility of our computational workflows was not assessed)

Line 36: “reproduction” was replaced by “replication” (we meant reproduction in the generic sense, but this idea is more clearly conveyed by the term replication as defined in the NASEM report)

Line 42: “reproduction” was replaced by “replication or reproduction” (access to scientific instruments, materials and software is necessary both for computational reproducibility or for attempting a complete replication of a previous study)

Line 50: “reproducibility of published results” was replaced by “reproducibility of published results or replication attempts” (both computational reproducibility using the same tools and data, and independent replication attempts can benefit from having access to the same tools used in the original studies)

Line 253: “reproducible and repeatable measurements” was replaced by “robust measurements” (the terms used in the paper cited to assess the quality of the measurements were not in line with the NASEM terminology, so they were replaced with a more general term)

Line 271: “reproducibility” was left unchanged (it refers to computational reproducibility being hindered by lack of data sharing standards)

Line 292: “reproducible” was removed (same reason as in the Abstract)

Reviewer's comment: 5. Line 44; if the user makes improvements to open source software, as described here, does that contribute to (or detract from) its benefits for others?

Authors' response: This sentence refers only to the freedoms conferred to the user when software is openly licensed. Any changes to a software can only impact other users as long as the modified version is further distributed. This is the same for proprietary or closed-source software, the difference being that in this case only the original developer can (legally) make those changes. In recent years, with the advent of subscription cloud platforms and software-as-a-service, the final user is usually not aware of software changes or updates, since they function as a “black box”. In the case of open-source software, the “risk” of many people being able to make changes is accompanied by the possibility of inspecting the code and effectively checking those changes against previous versions. We have not included any additional comments in the manuscript text, since these concepts apply to all openly licensed works and we therefore believe it is out of the scope of our paper to discuss it more deeply. The reader can refer to the literature already cited [10-16] for further details on this topic.

Reviewer's comment: 6. Lines 100-102; the statement that the hardware which is not open-hardware can be “readily replaced” doesn’t make sense in the context of the study; if the various commercial components such as Windows computers are convenient, why not commercial drones?

Authors' response: In contrast with open source software, open source hardware is virtually impossible to be 100% open source, and therefore uses the concept of “available components” (https://ohwr.org/project/cernohl/wikis/faq#q-what-are-available-component)s or “readily-available components and materials” (https://www.oshwa.org/definition/). The OSDT components that are referred to in lines 102-103 are those that can effectively be replaced without significantly altering the technical specifications of the system. It is not expected that the capabilities of the system will be significantly altered by using a different computer, smartphone or radio transmitter (since the drone flies autonomously), as long as they are compatible. This might not be the case for the camera, for which different brands or models could have widely varied capabilities, and therefore we suggest the use of a specific brand. 

A commercial drone could indeed be used as a complete replacement, but this generally also requires the use of a dedicated closed-source software, has limited customization options, etc. The aim of our work was to reduce the use of proprietary technologies as much as possible, therefore maximizing the user freedoms and the advantages of open tools.

Reviewer's comment: 7. P. 6; when referencing “GPS”, this would typically refer to Global Positioning System which is a United States platform; wouldn’t this benefit from a more generic reference, given international scope?

Authors' response: This was corrected as suggested:

Table 1: “GPS-guided flight” was replaced by “satellite-guided flight”

Table 1: “GPS coordinates” was replaced by “coordinates”

Line 94: “GPS capability” was replaced by “satellite navigation capability”

Table 2: “GPS recorder” was replaced by “Location recorder app”

Line 111: “GPS” was replaced by “GNSS (Global Navigation Satellite System)”

Line 142: “GPS location data” was replaced by “location data”

Line 193: “GPS logging app” was replaced by “location recorder app”

Reviewer's comment: 8. Line 185; I doubt Sentinel 2 imagery can be used to “validate” drone imagery that has a spatial resolution of 2 x 2 cm; these are very different sources and it is not clear what is meant by “validate”; I would strongly suggest including these efforts as a comparison but not as a validation

9. The use of the handheld sensor is helpful, though still a comparison rather than a validation; I believe the latter would require a far more sophisticated process which draws deeply upon literature regarding error expression and propagation as well as uncertainty

Authors' response: The objective in both cases was indeed to provide a comparison and not a validation, and it is referred to as such many times throughout the manuscript (lines 28, 203, 215, 223, 227, 248, 289). The term validation was wrongly included in this sentence, and has been corrected (line 188). 

Reviewer's comment: 10. You can reference literature that has already done extensive work on low altitude drone imaging using commercial sensors, and its comparisons with reference data; the fact that you are using an open hardware drone should not change the fundamental process of low altitude aerial image streams

Authors' response: In the paragraph in lines 247-258, our results are discussed in relation to some of the (indeed extensive) literature on UAV aerial imaging, and particularly the comparison of similar sensors (RGB cameras) to other sensors (multispectral cameras and LiDAR systems). In the revised version we have included additional references (line 255) to studies which have compared RGB cameras to multiple sensors, i.e., Micasense RedEdge-M camera [54], SlantRange 3p sensor [53], MicaSense RedEdge 3, Parrot Sequoia and Sentera cameras [55], Parrot Sequoia camera, Tec5 HandySpec Field spectrometer [52]. As pointed out by the Reviewer, our work should only aspire to provide a comparison to reference instruments, since a true validation would indeed require a sophisticated procedure which is beyond the scope of our paper.

---

Review Comments to the Author - Reviewer #2

Reviewer's comment: Thanks for the opportunity to review this paper, I really enjoyed reading it an learning about your work! The paper is well written and provides documentation of interesting science that is well aligned with my area of interest. I have attached my comments in the document and hopefully they prove useful for you in a minor revision.

Authors' response: We appreciate the reviewer’s comments and valuable suggestions. All the in-line comments in the manuscript have been addressed, as detailed below.

Reviewer's comment: I also suggest that the figure quality needs to be increased - they are quite blurry on my screen.

Authors' response: The figures are shown in low resolution in the PDF, but are uploaded in the required resolution in the editorial platform.

Reviewer's comments in the document:

Line 48: Comment: Might be worth noting that sometimes a disadvantage is that often open hardware and software is developed through a grant or similar and has no business model to maintain, sustain, or further develop [...]

Authors' response: We agree with the concerns about the sustainability of open source projects. We have added a paragraph on this issue in the Discussion section (lines 259-267).

Line 69: “The objective of this work was to address this question by collecting, curating, organizing and testing…” - Comment: processing? analysing?

Authors' response: We kept this phrase unchanged, since we consider that the processing and analysis of the data is included in the “testing” step

Line 108: Comment: I like the name

Authors' response: Thanks!

Line 122: “...with approximately 30% of battery capacity remaining to return to the launch site and land safely.” - Comment: Caveat that this is based on how far away (horizontally and vertically) the drone is from the home point, noting that there should be 20% battery remaining on landing

Authors' response: We agree that the time necessary to return to the launch site and land should be considered in the total flight time, and that there should be at least 20% battery remaining after that. We tried to make a conservative estimate considering 30% battery remaining, but it is true that our phrasing implied that that “extra” 10% would be sufficient for returning to the home point and landing in all cases, which is not correct. We therefore modified the sentence so as to avoid this (line 124).

Line 144: “This process is called geotagging … and can be performed using the Mission Planner ground station software.” - Comment: as long as you remember to set the time properly on the camera

Authors' response: This is indeed an important thing to consider, and as such it is included in the “usage guide” of the OSDT. Also, since it is something that is easy to forget, an alternative is to measure the time offset afterwards, and use that offset in the geotagging tool.

Line 202: Comment: I'm missing in this paragraph the height of the sensor from the feature, it's FOV, and therefore the 'footprint' of the measurement. Also maybe something about how the measurements were geo-located?

Authors' response: The missing information was included (lines 211-214)

Lines 202-208: [various corrections]

Authors' response: The text was corrected as suggested

Line 214: Comment: Would be good to note the spatial accuracy of the GPS and how you are effectively tying 2cm (GSD from drone image) to a satellite image and field data measurements when presumably there could be meters discrepancy? I understand that you can't resolve for this within the scope of the study, but would like to see recognition of this issue

Authors' response: We included a sentence which warns about this possible discrepancy in the case of the drone-satellite comparison (lines 205-208). In the case of the handheld sensor-drone comparison, a visible physical mark combined with ground measurements and the aid of the crop rows was used to aid in the correspondence (lines 211-213).

Lines 224-225: Comment: why not potentially commercial grade? I'm guessing here that you're imagining 'research grade' is lower level - but no reason why your results can't demonstrate value to the commercial sector? If the results are comparable to proprietary systems, why not?!

Authors' response: We agree about the possibility of commercial applications, but the scope of our work and the toolkit is scientific applications. We did not imply either lower or higher level, but requirements that are specific to scientific work (e.g., not only being able to provide a valid measurement but also being transparent about the algorithms that lead to that result). We nevertheless included a small comment to suggest the possibility of commercial applications (line 237).

Line 234: I think that it's often the commercial sector that perhaps hints at the 'quality' or lack thereof of open source (perhaps a reasonable tactic to keep themselves in business, which of course is necessary for them too), but the reliability side of things can be a real issue (see my point in the intro.

Authors' response: As indicated in the response to the first comment, a paragraph on this topic was added to the Discussion section (lines 259-267).

Line 239: Comment: I would also argue that in many cases, the high spatial resolution is just as, if not more important the gaining an extra couple of bands. Multispec cameras are notoriously low in spatial resolution, and people who don't understand this trade off relationship are often left disappointed with their purchases. The Phantom multispec for example is 2MP vs 20MP for their standard RGB (I know you are going with open source, but just for comparison here).

Another thought actually could be to hack a canon or similar to remove the NIR filter and then put filters back on in the wavelengths of interest - that would be more in your flavour and keep the spatial resolution of the sensor.

Note that I suggest the above as comments to consider adding in the discussion text, not for re-doing or adding to the scope of your current paper

Authors' response: We agree on the advantages of the higher spatial resolution of RGB over multipectral cameras, as clearly explained by the Reviewer. We added a comment and a reference to a paper where this is discussed in more detail (line 255). We also added references about the possibility of using modified RGB cameras for NIR imaging (lines 145-147).

Line 244: “...little attention has been paid to the management…” - Comment: and storage

Authors' response: We included the text as suggested.

---

## [Decision Letter · Decision Letter 1]

20 Mar 2023

PONE-D-22-24873R1Open Science Drone Toolkit: open source hardware and software for aerial data capturePLOS ONE

Dear Dr. Pereyra Irujo,

Thank you for submitting your manuscript to PLOS ONE. After careful consideration, we feel that it has merit but does not fully meet PLOS ONE’s publication criteria as it currently stands. Therefore, we invite you to submit a revised version of the manuscript that addresses the points raised during the review process. Reviewers higlighted the fact that authors have very carefully considered the reviewers comments. This is an important study involving a significant effort. There is still a minor recommandation to be addressed before publication.

We look forward to receiving your revised manuscript.

Kind regards,

Vona Méléder, Ph.D.

Academic Editor

PLOS ONE

Journal Requirements:

Reviewers' comments:

Reviewer's Responses to Questions

**Comments to the Author**

1. If the authors have adequately addressed your comments raised in a previous round of review and you feel that this manuscript is now acceptable for publication, you may indicate that here to bypass the “Comments to the Author” section, enter your conflict of interest statement in the “Confidential to Editor” section, and submit your "Accept" recommendation.

Reviewer #1: All comments have been addressed

2. Is the manuscript technically sound, and do the data support the conclusions?

Reviewer #1: Yes

3. Has the statistical analysis been performed appropriately and rigorously? 

Reviewer #1: N/A

4. Have the authors made all data underlying the findings in their manuscript fully available?

Reviewer #1: Yes

5. Is the manuscript presented in an intelligible fashion and written in standard English?

Reviewer #1: Yes

6. Review Comments to the Author

Reviewer #1: Thank you for the opportunity to review this new version of the manuscript. You have carefully addressed the reviewer comments. I am pleased to recommend publication with the following minor recommendation:

I still get a sense that the comparative data are being presented as "reference" and as an "evaluation" rather than simply a comparison; the above words probably require a different approach, which I don't believe is necessary; I suggest changing these words to "comparison" or to a close synonym.

7. PLOS authors have the option to publish the peer review history of their article (what does this mean?). If published, this will include your full peer review and any attached files.

Reviewer #1: No

---

## [Author Response · Author response to Decision Letter 1]

22 Mar 2023

Review Comments to the Author - Reviewer #1:

"Thank you for the opportunity to review this new version of the manuscript. You have carefully addressed the reviewer comments. I am pleased to recommend publication with the following minor recommendation:

I still get a sense that the comparative data are being presented as "reference" and as an "evaluation" rather than simply a comparison; the above words probably require a different approach, which I don't believe is necessary; I suggest changing these words to "comparison" or to a close synonym."

Response: We appreciate the Reviewer’s recommendation for publication. As suggested, the words “reference” and “evaluation” were deleted or replaced with “comparison” or similar words, as follows:

-Lines 28-30 (Abstract): “Performance was evaluated by…” was replaced by “Data obtained with this toolkit … was compared to…”, and “both reference instruments” was changed to simply “both instruments”.

-Line 188: “two reference sources” was changed to “two sources”.

-Line 248: “both reference instruments” was replaced by “both instruments”.

-Lines 288-290: Same changes as in the Abstract

---

## [Editor Report · Decision Letter 2]

27 Mar 2023

Open Science Drone Toolkit: open source hardware and software for aerial data capture

PONE-D-22-24873R2

Dear Dr. Pereyra Irujo,

We’re pleased to inform you that your manuscript has been judged scientifically suitable for publication and will be formally accepted for publication once it meets all outstanding technical requirements.

Kind regards,

Vona Méléder, Ph.D.

Academic Editor

PLOS ONE
---

## [Editor Report · Acceptance letter]

29 Mar 2023

PONE-D-22-24873R2 

Open Science Drone Toolkit: open source hardware and software for aerial data capture 

Dear Dr. Pereyra Irujo:

I'm pleased to inform you that your manuscript has been deemed suitable for publication in PLOS ONE. Congratulations! Your manuscript is now with our production department. 

Kind regards, 

on behalf of

Dr. Vona Méléder 

Academic Editor

PLOS ONE